

# Transcriptome analysis reveals the mechanism of improving erect-plant-type peanut yield by single-seeding precision sowing

Sha Yang[1,2,*], Jialei Zhang[1,2,*], Yun Geng[1,2], Zhaohui Tang[1,2], Jianguo Wang[1,2], Feng Guo[1,2], Jingjing Meng[1,2], Quan Wang[1,3], Shubo Wan[4] and Xinguo Li[1,2]

[1] Biotechnology Research Center, Shandong Academy of Agricultural Sciences, Ji'nan, China
[2] Scientific observation and experiment station of crop cultivation in east China, Ministry of Agriculture and Rural Affairs, Dongying, China
[3] College of Life Sciences, Shandong Normal University, Ji'nan, China
[4] Shandong Academy of Agricultural Sciences/Shandong Provincial Key Laboratory of Crop Genetic Improvement, Ecology and Physiology, Ji'nan, China
* These authors contributed equally to this work.

Corresponding authors
Shubo Wan, wansb@saas.ac.cn
Xinguo Li, xinguol@163.com

## ABSTRACT

**Background:** In China, double-seed (DS) sowing (i.e., sowing two seeds per hole) has been conventionally performed towards the erect-plant-type peanuts to increase the low germination rate due to poor seed preservation conditions. However, the corresponding within-hole plant competition usually limits the subsequent plant growth and the final yield. We developed a high-yield cultivation system of single-seed (SS) precision sowing to solve this paradox, saving 20% of seeds and increasing yields by more than 10% relative to the conventional DS sowing.
**Methods:** To explore the mechanisms of these two different cropping patterns in peanut yields, we conducted transcriptomic and physiological comparisons in the seeding plant leaf and root tissues between SS precision sowing and standard DS sowing treatments.
**Results:** After assembly, each library contained an average of 43 million reads and generated a total of 523,800, 338 clean reads. After GO and Kyoto Encyclopedia of Genes and Genomes pathway analysis, we found the key genes for biotic and abiotic stress showed higher expression in roots of plants grown under the SS precision sowing treatment, including genes encoding disease resistance, oxidation-reduction, hormone related, and stress response transcription factors and signaling regulation proteins. In particular, the resveratrol synthesis genes related to stress and disease resistance appeared induced in roots under the SS sowing treatment.
**Conclusion:** These data indicated that *Aspergillus flavus* resistance and stress tolerance in roots under SS precision sowing were enhanced compared with roots under the DS sowing treatment. This work benefits the development of underground pods and thus increasing peanut yields.

## INTRODUCTION

Before the 1960s, agricultural production in China was labor-intensive due to the lack of modern farming conditions. Thus, erect-plant-type peanuts, which are easier for manual labor relative to prostrate-plant-type peanuts, were successfully bred and cultivated widely in China. Because the seed germination rate of peanut is usually unable to meet the necessary high field emergence percentage owing to uneven seed quality, conventionally, the sowing patterns with double-seed (DS) and multi-seed in one hole were widely adopted by farmers as the main planting patterns, which generally leads to two issues: a requirement of a large number of seeds and poor population quality. The total amount of peanut seeds per year is 1.5 million tons, accounting for about 10% of the country's total peanut yield. Poor population quality is mainly attributed to the increased competition under DS sowing, which hinders the increases of peanut yield (*Zhang et al., 2020*).

Our team introduced the principle of competitive exclusion, proposed the technical idea of "single-seed (SS) precision sowing of erect-plant-type peanuts, robust individual, optimized population", and created a high-yield cultivation technique of SS precision sowing. SS precision sowing can alleviate inter-plant competition, save 20% of seeds used for planting, and increase yields by more than 10% in common fields relative to DS precision sowing (*Liang et al., 2020*). However, the molecular mechanisms regulating individual development under SS are still unknown.

Crop yield is usually affected by biotic and abiotic stress. In recent years, biotic stress research has mainly focused on disease resistance in wheat, rice, maize, and soybean. It has been reported that *Fhb1* was associated with resistance to wheat scab and has important breeding potential (*Rawat et al., 2016*). Two grape stilbene synthase (STS) genes, *VST1* and *VST2*, were transformed into tobacco for the first time, which improved tobacco resistance to *Botrytis cinerea* (*Hain et al., 1993*). "MutRenSeq", proposed by a British research team, is an efficient method to discover disease-resistant genes in crops (*Steuernagel et al., 2016*). These advances lay a solid foundation for the further study of crop disease resistance.

Aflatoxin contamination caused by *Aspergillus flavus* is a major obstacle to the development of peanuts (*Fajardo et al., 1994*). It was indicated that the resveratrol (Res) content could be significantly induced when *Aspergillus flavus* infects the peanuts seeds; and resistance of peanut to *Aspergillus flavus* infection can be conferred by increasing the content of Res in peanut seeds as well as the speed of its synthesis (*Tian et al., 2008*). Res, known as 3,4,5-trihydroxy-stilbene, was first isolated from *Veratrum grandiflorum* root in 1940 (*Takaoka, 1940*). At present, Res has been identified in many different plant species, and recent research is mainly focusing on grapes (*Wang et al., 2010*), peanuts (*Tang et al., 2010*), and a few other species. Grape pericarps are the main grape tissue in which the compound is synthesized. The content of Res in peanut pericarps is higher than that in peanut kernels (*Sanders, McMichael & Hendrix, 2000*). Accumulated of Res, a non-flavonoid polyphenol, could improve abiotic stress resistance. Although there has remained a query of the specific mechanisms of Res's anti-stress abilities, a recent report

showed that Res resistance to ROS is one aspect involved in the stress-resistant processes (*Zheng et al., 2015*).

Abiotic stresses, including drought, salt and extreme temperature, remain the most challenging frontier in the field of plant abiotic interaction research. Research on abiotic stress focuses on identifying primary sensors and essential stress-resistant genes that respond to abiotic stresses (*Zhu, 2016*). Significant progress has been made in the past few years. The cold-tolerant rice protein COLD1 has been identified to mediate extracellular $Ca^{2+}$ influx and net cytosolic $Ca^{2+}$ concentration in response to chilling stress (*Ma et al., 2015*). The basic leucine zipper bZIP and NAC transcription factors (TF) may sense or contribute tolerance to salt stress through their interaction with corresponding proteins (*Liu & Howell, 2016*). The chloroplast is the main site of the production of reactive oxygen species (ROS), including hydrogen peroxide, superoxide anions, singlet oxygen and hydroxyl radicals (*Mignolet-Spruyt et al., 2016*). Various abiotic stresses, particularly high light stress, aggravate ROS production, destroying ROS-scavenging systems and generating various secondary messengers. The protective mechanisms that scavenge ROS in plants can be divided into two categories: enzymatic (e.g., superoxide dismutase (SOD), catalase (CAT) and peroxidase (POD)) mechanisms and non-enzymatic (e.g., glutathione, mannitol and flavonoids) mechanisms. Improving the content of these substances in plants by genetic engineering can effectively remove excessive ROS in plants and thus improve stress tolerance. Whether the yield advantage of SS precision sowing is related to the stress resistance genes needs to be studied.

With the development of high-throughput sequencing technologies, more attention is being paid to combining genomic methods such as genome sequencing with transcriptome, proteome and metabolome analyses in order to reveal the molecular mechanism of life science research. This integrative research offers synergies that can uncover the molecular mechanism of different phenomena in crop production. The publication of reference transcripts for *Arachis_duranensis* (*Chopra et al., 2014*) facilitated a better understanding of agronomically important phenomena and genetic improvement of peanuts. This study used these techniques mentioned above to reveal the mechanism attributing to the higher peanuts yield by SS precision sowing.

# MATERIALS AND METHODS

## Plant materials and growth conditions

Peanut (*Arachis hypogaea* L.) cultivar "Huayu 22" (provided by the Shandong Peanut Research Institute, China), a large-grain peanut cultivar, was used in this study. Peanut seeds were cultivated under field pot conditions at the Shandong Academy of Agricultural Sciences Station (117°5′ E, 36°43′ N), Ji'nan, China. The soil is sandy loam, containing 1.1% organic matter (W/W), 82.7 mg $kg^{-1}$ alkali-hydrolyzed nitrogen, 36.2 mg $kg^{-1}$ available phosphorus, 94.5 mg $kg^{-1}$ available potassium and 14.9 g $kg^{-1}$ exchangeable calcium. The SS sowing treatment consisted of 237,000 holes per hectare with 1 grain per

hole, while the DS sowing consisted of 138,000 holes per hectare with two grains per hole as the control; the distances between two adjacent holes were 10.5 cm and 18 cm in the SS and DS treatments, respectively. Three biological replicates were used in this study. The mixed tissues were used in transcriptome sequencing.

## RNA extraction and cDNA library construction

When the peanut plants reached the seedling stage, the top third leaf and the whole root systems were collected for RNA extraction and library construction. Total RNA was isolated from the leaves and roots respectively, using the total RNAiso Reagent (TaKaRa, Dalian, China) according to the manufacturer's instructions. The quality and purity of RNA samples were detected using the Agilent 2100 Bioanalyzer platform and Agilent RNA 6000 Nano Kit (Agilent, Santa Clara, CA, USA). Qualified RNA samples were used to isolate mRNA with the oligo (dT) method and the mRNAs were fragmented. Then, first-strand cDNA and second strand cDNA were synthesized. Next, the purified cDNA fragments were linked with adapters, and sequencing was performed by a commercial service provider (BGI Tech, Shenzhen, China). The sequenced reads that contained adaptor sequences, low-quality bases, and high contents of unknown bases (i.e., N calls) were removed before downstream analyses. After read filtering, clean reads were mapped to the *Arachis_duranensis* reference genome (BioSample: SAMN02982871, BioProject: PRJNA258023) using Hierarchical Indexing for Spliced Alignment of Transcripts (HISAT). The sequence data can be located from https://www.ncbi.nlm.nih.gov/assembly/ GCF_000817695.2/. Then, Genome Analysis Toolkit (GATK) was used to call single SNPs and INDELs (*McKenna et al., 2010*) and RNA-Seq by Expectation-Maximization (RSEM) was used to calculate gene expression levels for each sample, which were then determined using the fragments per kb per million (FPKM) mapped fragments method developed by *Li & Dewey (2011)*. DEGseq algorithms were used to assess the DEGs. Based on the gene expression level, DEG was identified between samples or groups. Gene Ontology (GO) was performed classification and Kyoto Encyclopedia of Genes and Genomes (KEGG) pathway classification based on these DEGs. The GO framework includes three ontologies: molecular biological function, cellular component and biological process. KEGG is a database resource for understanding high-level functions and utilities of the biological system, such as the cell, the organism and the ecosystem, from molecular-level information, especially large-scale molecular datasets generated by genome sequencing and other high-throughput experimental technologies. Co-expression analysis was performed using the WGCNA R package according to the methods detailed by previously published studies (*Gao et al., 2018*; *Song et al., 2018*).

## Yield composition per plant and measurement

At maturity, 10 representative plants were selected to investigate the number of pods, full pods and double kernels, as well as pod weight per plant. The pods and plants were dried to constant weights, and the economic coefficient was calculated as follows: economic coefficient = pod dry weight/(plant dry weight + pod dry weight).

## Antioxidant enzyme activities and root activity

Leaf tissues (0.5 g) were ground up with phosphate buffer (pH 7.8) containing 0.1 mM EDTA and 1% (g ml$^{-1}$) PVP and centrifuged at 4 °C for 10 min. Then, the supernatants were used as enzyme extracts. The nitroblue tetrazole (NBT) method was used to detect SOD activity. The absorbance value was determined at a 560 mm wavelength after the reaction; 50% inhibition of NBT reduction was regarded as the enzyme activity unit (U), expressed as U g$^{-1}$ FW. The guaiacol method was used to determine the POD activity; the increase in $OD_{470}$ per min was used as the enzyme activity unit (U), expressed as $\Delta$470 g$^{-1}$. CAT activity was also determined by the guaiacol method. The decreased in $OD_{240}$ per min was used as the unit of enzyme activity (U) and expressed as mg·g$^{-1}$·min$^{-1}$ (*Giannopolitis & Ries, 1977*; *Aebi, 1984*; *Jimenez et al., 1997*). Root activity was measured by the TTC method; 0.5 g of root tip samples were revolved in a mixture of 0.4% TTC and phosphoric acid buffer solution and kept in the dark for 1–3 h at 37 °C. After that, 2 ml of 1 mol L$^{-1}$ sulfuric acid was added to stop the reaction. Simultaneously, a blank experiment was conducted. The root samples and sulfuric acid were added first, and 10 min later, the other reagents were added. The operation was the same as that described above. The roots were removed, dried, and ground together with 3–4 ml of quartz sand and ethyl acetate in a mortar. The supernatants were transferred to a test tube, and the residue was washed with a small amount of ethyl acetate two or three times, with all ethyl acetate collected into the test tube. Finally, ethyl acetate was added to bring the total volume up to 10 ml. Blank experiments were used as the control, and the OD value of the reaction solution was measured at $\lambda$ = 485 nm.

## qRT-PCR verification analysis

Leaves and roots of plants grown under the SS and DS treatments, as in the RNA-seq experiment, were selected randomly and analyzed using qRT-PCR. The qRT-PCR amplification instrument (ABI 7500 fast; Applied Biosystems, Foster City, CA, USA) was used to amplify the related genes with SYBR Premix Ex Taq$^{TM}$ (TaKaRa, Nugegoda, Sri Lanka) following the manufacturer's instructions (*Yang et al., 2013*). The relative gene expression was calculated using the $2^{-\Delta\Delta Ct}$ method described by *Livak & Schmittgen (2001)*. For log$_2$-transformed FPKM values, each selected gene's maximum expression level was considered to be 100, and the expression levels of the other genes were transformed accordingly (*Zhang et al., 2016*).

## Determination of Res content

The Res extraction processes were carried out according to the method detailed by *Tang et al. (2010)* with minor modification; 0.25 g of leaves or roots were collected and ground into a fine powder, which was dissolved with 20 ml of 95% ethanol in a flask and oscillated at room temperature overnight. After centrifugation at 8,000 rpm for 10 min, the supernatants were subjected to HPLC analysis. The extracts were filtered through a 0.45-μm membrane and then tested with the Rigol L3000 HPLC system (Rigol Technologies, Beaverton, OR, USA) using a Kromasil C18 reversed phase column (4 μm, 250 mm × 4.6 mm; Kromasil, Bohus, Sweden) at room temperature. Mobile phase A

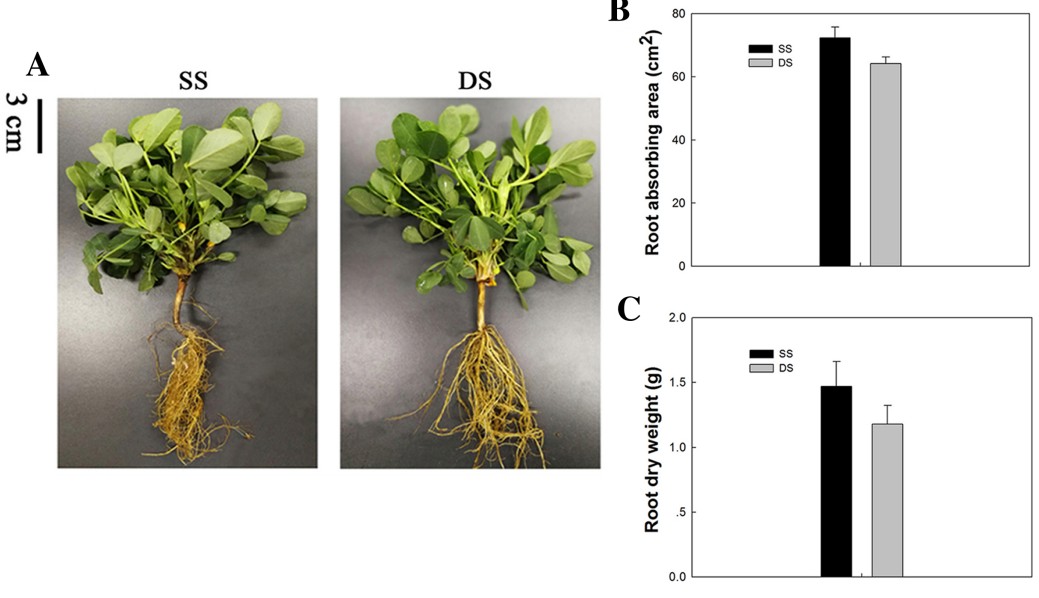

**Figure 1 Phenotypical comparisons of peanut in SS and DS sowing treatment.** (A) The phenotypes, (B) absorbing area of root in SS and DS and (C) Dry weight of root in SS and DS.

consisted of 0.1% phosphoric acid–$H_2O$, while phase B consisted of acetonitrile (A: B = 70:30, flow rate 1 ml min$^{-1}$); the determination wavelength was 280 nm.

## RESULTS

### Transcriptome sequencing and gene analysis

The peanut plants grown under different planting patterns differed little in their above-ground portions, while great differences were observed between their root systems. The basis of high and stable crop yields is high biological yields, which depends largely on the development of root systems. The capillary lateral root is the main type of root exhibited under SS. Thus, the capillary lateral root length and its proportion within the root system were promoted accordingly, thereby increasing the total root length and root absorption area of peanut plants. This phenomenon was verified by the mean absorbing area and dry weight of root tissue observed under SS (Fig. 1).

A total of 12 cDNA libraries were constructed from leaf and root tissues of peanut plants grown under SS and DS sowing treatments. All 12 samples were sequenced on the Illumina HiSeq Platform in total, generating about 6.55 Gb per sample. The reads that contained adaptor sequences of low-quality or had high contents of unknown base (i.e., N) calls were removed, resulting in a total of 523,800,338 clean reads that were acquired, with an average of 43 million reads per libraries (Table 1). Approximately 77% and 80% of reads from leaf and root tissues were mapped to the reference genome and 36,778 genes were identified, of which 34,529 are known genes and 2,322 are novel genes. The RNA-seq sequencing data for the present work has been uploaded in NCBI Sequence Read Archive under BioProjects, PRJNA497502 (SRA: SRP166140).

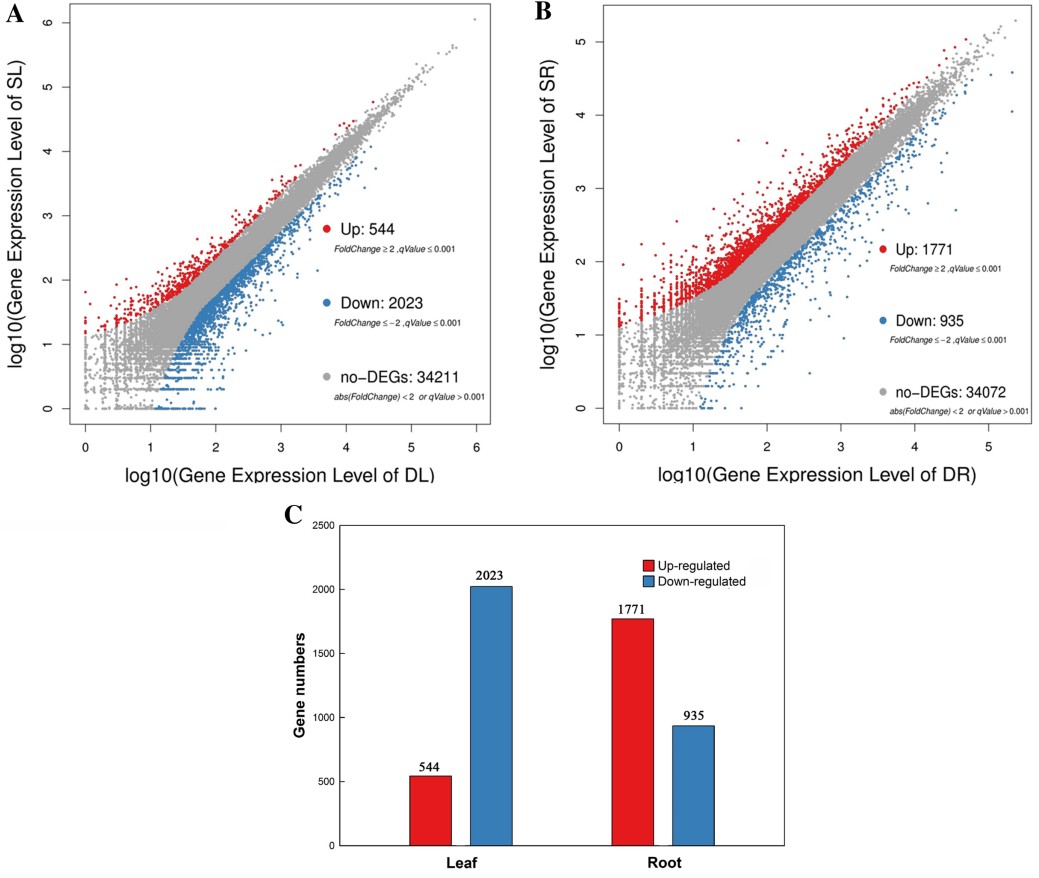

**Figure 2 Scatter plot of DEGs in leaf and root between SS and DS treatments.** (A) MA-Plot distribution of DEGs in leaves. (B) MA-Plot distribution of DEGs in roots. (C) Statistical chart of DGEs in leaves and roots.

    Single nucleotide polymorphisms (SNP), which include transitions or transversions of single bases, refer to the difference of a single nucleotide (A, T, C, or G) between homologous DNA sequences, a key element of the diversity of genomes among species or individuals. There were more transitions than transversions among all the samples. Among SNPs, transition, transversion, A–G and C–T were the most abundant terms (Fig. S1). Insertion–Deletion (INDEL) refers to the insertion or deletion of the small fragments (one or more, less than 50 bp) that occur in a sample relative to the reference genome. Most INDELs occurred in exonic and intronic sequences, with the proportion differing among samples (Fig. S2). Based on the results of SNP, INDEL and gene expression, our study was presented in the form of a ring diagram with Circos software (Fig. S3).

## Differentially expressed gene detection
A total of 2,567 and 2,706 differentially expressed genes (DEGs) were identified from leaves to roots, respectively. Under SS sowing treatment, 544 and 1,771 genes were expressed at a higher level, while 2,023 and 935 genes were expressed at lower levels compared with the DS sowing treatment in leaves and roots, respectively (Fig. 2). The expression patterns of the majority of DEGs differed between roots and leaves.
**Table 1 Summary of read numbers from leaves and roots.**

| Samples | Reads in leaf | | | | | | Reads in root | | | | | |
|---|---|---|---|---|---|---|---|---|---|---|---|---|
| | SL1 | SL2 | SL3 | DL1 | DL2 | DL3 | SR1 | SR2 | SR3 | DR1 | DR2 | DR3 |
| Total clean reads | 45,001,686 | 44,475,256 | 44,262,552 | 42,022,054 | 42,294,220 | 42,570,982 | 42,569,308 | 42,180,566 | 44,533,150 | 44,784,894 | 44,608,344 | 44,497,326 |
| Total Mapping Ratio | 75.27% | 74.87% | 74.88% | 78.26% | 80.52% | 80.60% | 80.73% | 80.88% | 82.71% | 78.44% | 78.15% | 80.95% |
| Unique match Ratio | 55.15% | 53.51% | 54.34% | 49.41% | 51.29% | 54.53% | 51.93% | 52.18% | 53.61% | 46.73% | 43.23% | 52.30% |
| Unmapped reads | 24.73% | 25.13% | 25.12% | 21.74% | 19.48% | 19.40% | 19.27% | 19.12% | 17.29% | 21.56% | 21.85% | 19.05% |
| Novel transcript number | 13,152 | 12,615 | 13,149 | 13,218 | 13,030 | 12,895 | 13,139 | 13,136 | 13,316 | 13,021 | 12,911 | 12,909 |

**Note:**
SL, leaf in single-seed (SS) precision sowing; DL, leaf in double-seed (DS) sowing; SR, root in single-seed (SS) precision sowing; DR, root in double-seed (DS) sowing.

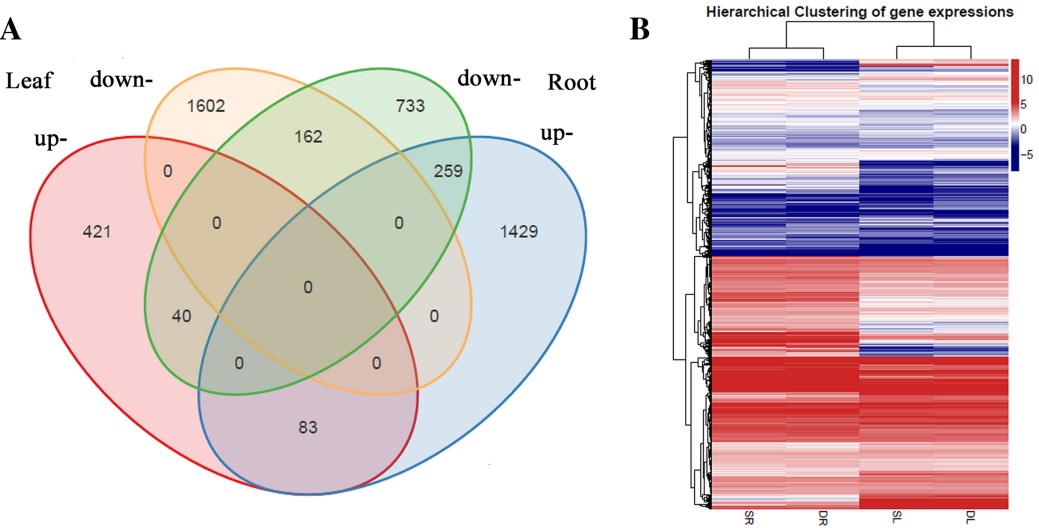

**Figure 3 Differentially expressed genes analysis in roots and leaves.** (A) Venn diagram demonstrated the common and specific differentially expressed genes (DEGs). (B) Heat map demonstrated the expression profile DEGs.

For example, only 83 (15.2%) of the 1,771 genes up-regulated in roots were also up-regulated in leaves (Fig. 3). Notably, some genes in roots and leaves even showed opposite expression patterns; for example, forty DEGs were up-regulated in leaves but down-regulated in roots. However, 245 DEGs in roots and leaves had the same expression trends, including 83 up-regulated DEGs and 162 down-regulated DEGs (Fig. 3).

## qRT-PCR verification of RNA-seq results

To verify the RNA-seq data, quantitative real-time PCR (qRT-PCR) was used to test the expression of 24 genes with different functional assignments (Figs. 4A and 4B). Among them, six and five genes were up-regulated under the SS sowing treatment in leaves and roots, respectively. UDP-glycosyltransferase, chalcone synthase and GPI

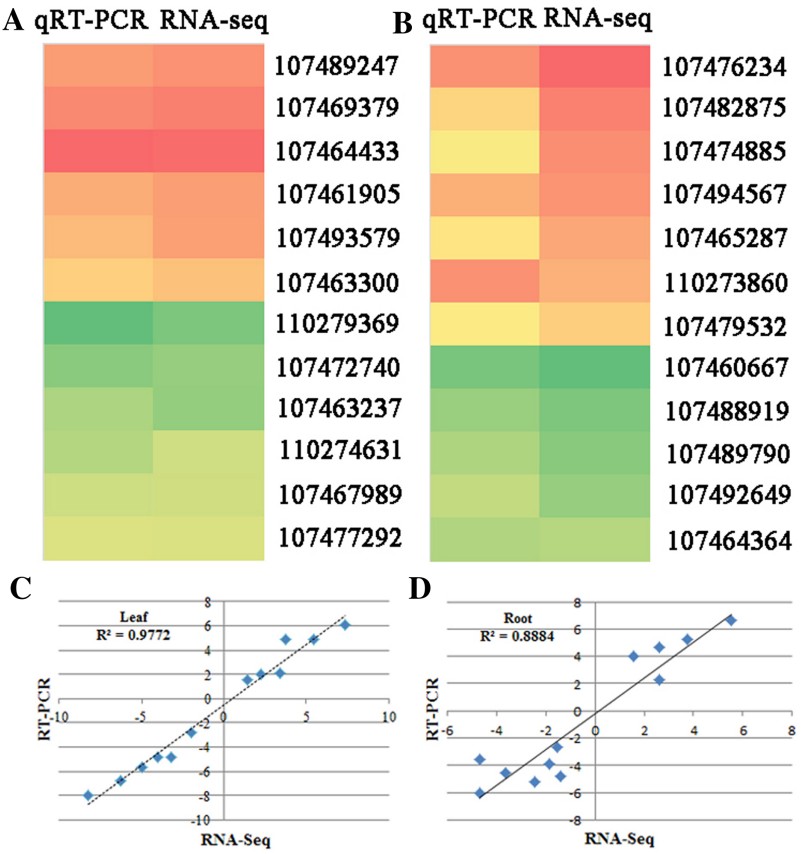

**Figure 4 qRT-PCR verification of randomly selected genes.** Heatmaps represent the expression of the 22 DEGs in (A) leaves and (B) roots between SS and DS treatments. For each heatmap, the FPKM values increased from green to red. The correlation of the fold change in (C) leaves and (D) roots analyzed by RNA-Seq (*x*-axis) with data obtained using qRT-PCR (*y*-axis).

mannosyltransferase were each encoded by one of these genes, while two genes encoding hypothetical proteins were also used for qRT-PCR detection. The other eleven selected genes were downregulated both in roots and leaves. These genes included one phospholipase, one seed linoleate 9S-lipoxygenase, one zinc finger SWIM domain-containing protein, and one pectinesterase. The RNA-seq data were consistent with the qRT-PCR results for these 24 genes. The correlation between the relative expression (log$_2$ SS/DS) estimated by RNA-seq and the qRT-PCR results were rather high in leaves ($R^2$ = 0.9772) and a little lower in roots ($R^2$ = 0.8884; Figs. 4C and 4D).

## Functional analysis of DEGs

Blast2GO was used to perform GO classification and functional enrichment (*Conesa et al., 2005*) on the identified DEGs. The sequences were categorized into 45 functional groups according to sequence homology. The main categories of biological process, molecular function, and cellular component were visualized employing WEGO (Fig. 5). In the biological process category, "metabolic process", "single-organism process" and "cellular

 

![PeerJ]

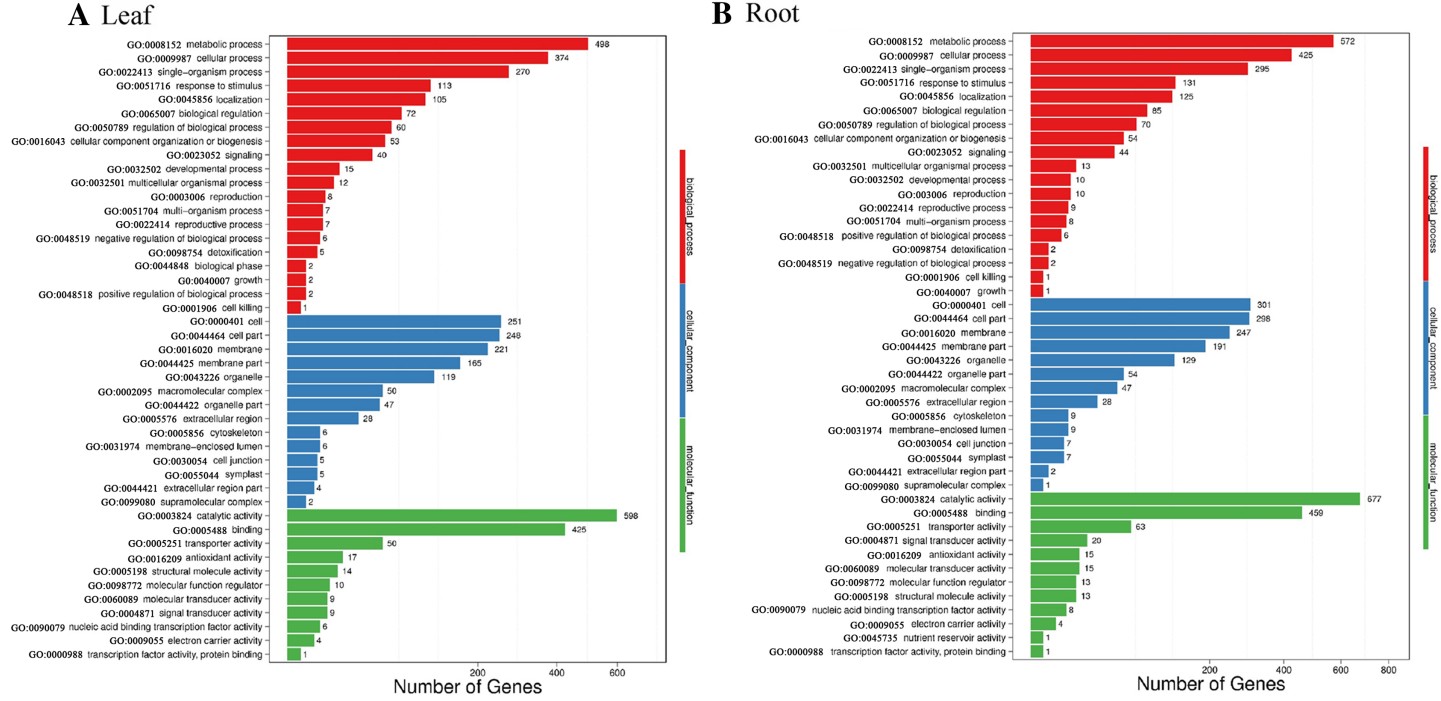

**Figure 5 GO annotation among DEGs identified in the leaf of SS treatment compared with DS treatment and root from the two different treatments.** The GO ID was listed alongside the textual annotations. (A) GO annotation among DEGs identified in the leaves between SS and DS treatments. (B) GO annotation among DEGs identified in the roots between SS and DS treatments.

process" terms were enriched, suggesting the metabolic activity is higher under SS sowing. In the cellular component category, "cell", "cell part" and "membrane" were the most abundant terms. The most enriched category was molecular function, with enrichment for "binding", "catalytic activity" and "transporter activity" in particular, suggesting a high level of metabolic activity changes under SS sowing (Fig. 5).

Kyoto Encyclopedia of Genes and Genomes pathway classification was performed to acquire biological information for understanding the regulatory networks and molecular mechanisms associated with the SS sowing treatment. DEGs were mainly enriched in the MAPK signaling pathway, glycerolipid metabolism, phenylalanine metabolism, sphingolipid metabolism, isoflavonoid biosynthesis, flavonoid biosynthesis and tryptophan metabolism in leaves. Meanwhile, phenylpropanoid biosynthesis, biosynthesis of secondary metabolites, MAPK signaling pathway, flavonoid biosynthesis, zeatin biosynthesis and flavone and flavonol biosynthesis were mainly enriched among DEGs in roots (Fig. S4). Interestingly, all of these pathways participated in biosynthesis associated with particular metabolic processes, suggesting that these processes were activated. Weighted gene correlation network analysis (WGCNA) is a systematic biological method for describing gene association pattern among different samples. Information about nearly 10,000 genes, corresponding to the genes with the greatest changes in expression, was used to identify gene sets of interest and analyze associations with phenotypes. In our study, the genes from all the samples were divided into modules for analysis, with twelve colors representing each of the different modules in Fig. 6.

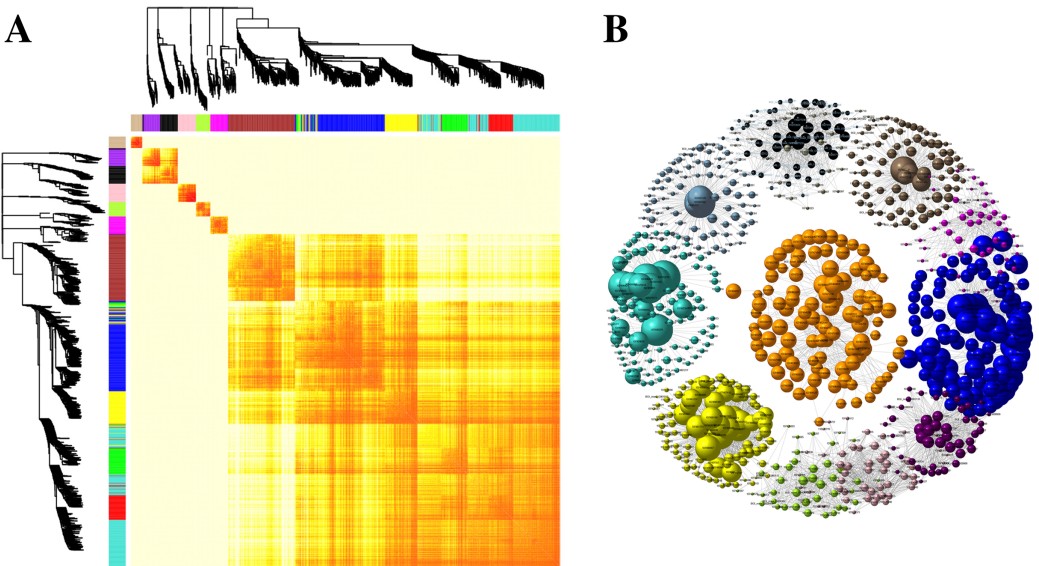

**Figure 6 Co-expression of DEGs between SS and DS treatments.** (A) Hierarchical of 10,000 genes based on topological overlap and visualization of gene modules with assigned colors. Branches in the hierarchical clustering dendrograms correspond to modules. Color-coded module membership is displayed in the colored bars below and to the right of the dendrograms. (B) Network of co-expressed modules. The colored clustering bars directly correspond to the module (color) designation for the clusters of genes.

## Changes of Res synthesis related pathway between cultivation techniques

In our study, a total of 20 Res synthesis-related genes were dramatically induced, while no Res synthesis-related genes were down-regulated in the roots under the SS sowing treatment. In contrast, a total of 10 DEGs related to Res synthesis were down-regulated in the leaves (Table 2). We also measured the Res content, and the results were in agreement with the gene expression levels in leaf and root tissues (Fig. 7). Accordingly, compared with the roots of plants grown under DS sowing, the higher Res content under SS precision sowing appeared to support the initiation of the defense response against toxins produced by *Aspergillus flavus*; pathogenesis-related proteins, TFs (e.g., WRKY, bZIP, ERF) and 4CL were up-regulated (Table 3).

## Key genes related to stress tolerance under SS

Transcription factors, which bind to the *cis*-elements upstream of promoters, have been reported to orchestrate abiotic stress responses (*Joshi et al., 2016*). Under the SS precision sowing treatment, the higher expression of these TF genes (WRKY, MYB, bZIP, ERF) may improve stress tolerance compared with that observed in the DS treatment. In plants, redox processes also play an important role in stress tolerance. Several oxidoreductase genes encoding ascorbate oxidase, glutaredoxin, and cytochrome P450 monooxygenases (CYP) exhibited increased expression in roots under SS relative to DS. Specifically, compared with DS, about thirty genes encoding CYP were up-regulated under SS (Table 3). In addition, four genes encoding l-ascorbate oxidase homologs, which take part

**Table 2 Analysis of resveratrol synthesis related genes in leaves and roots.**

| | Annotation | $\log_2^{(SL/DL)}$ | Gene expression change |
|---|---|---|---|
| Leaf | Resveratrol synthase | −1.43 | Down-regulated |
| | Resveratrol synthase | −1.92 | Down-regulated |
| | Resveratrol synthase | −2.03 | Down-regulated |
| | Resveratrol synthase | −2.31 | Down-regulated |
| | Resveratrol synthase | −2.66 | Down-regulated |
| | Resveratrol synthase | −2.77 | Down-regulated |
| | Resveratrol synthase | −3.36 | Down-regulated |
| | Resveratrol synthase | −6.66 | Down-regulated |
| | Stilbene synthase | −1.18 | Down-regulated |
| | Stilbene synthase | −1.38 | Down-regulated |
| | **Annotation** | **$\log_2^{(SR/DR)}$** | **Gene expression change** |
| Root | Resveratrol synthase | 2.67 | Up-regulated |
| | Resveratrol synthase | 2.59 | Up-regulated |
| | Resveratrol synthase | 2.37 | Up-regulated |
| | Resveratrol synthase | 2.27 | Up-regulated |
| | Resveratrol synthase | 2.24 | Up-regulated |
| | Resveratrol synthase | 2.13 | Up-regulated |
| | Resveratrol synthase | 2.12 | Up-regulated |
| | Resveratrol synthase | 2.05 | Up-regulated |
| | Resveratrol synthase | 1.86 | Up-regulated |
| | Resveratrol synthase | 1.80 | Up-regulated |
| | Resveratrol synthase | 1.68 | Up-regulated |
| | Resveratrol synthase | 1.67 | Up-regulated |
| | Resveratrol synthase | 1.61 | Up-regulated |
| | Resveratrol synthase | 1.56 | Up-regulated |
| | Resveratrol synthase | 1.55 | Up-regulated |
| | Resveratrol synthase | 1.47 | Up-regulated |
| | Resveratrol synthase | 1.34 | Up-regulated |
| | Resveratrol synthase | 1.27 | Up-regulated |
| | Stilbene synthase | 2.47 | Up-regulated |
| | Stilbene synthase | 2.08 | Up-regulated |
| | Stilbene synthase | 1.19 | Up-regulated |

**Note:**
The ratio was obtained from the transcriptome results. SL, leaf in single-seed (SS) precision sowing; DL, leaf in double-seed (DS) sowing; SR, root in single-seed (SS) precision sowing; DR, root in double-seed (DS) sowing.

in ascorbate recycling, were enriched in the SS sowing treatment. In plants, ascorbate contributes to improving tolerance against various stresses by regulating the levels of cellular $H_2O_2$ (*Ishikawa & Shigeoka, 2008*). Higher enzyme activities of POD, SOD, and CAT were also observed in roots under SS compared to that under DS (Table S1). Four genes encoding glutaredoxin, which might promote reducing disulfide bridges, were slightly induced under SS compared with DS (Table 3).

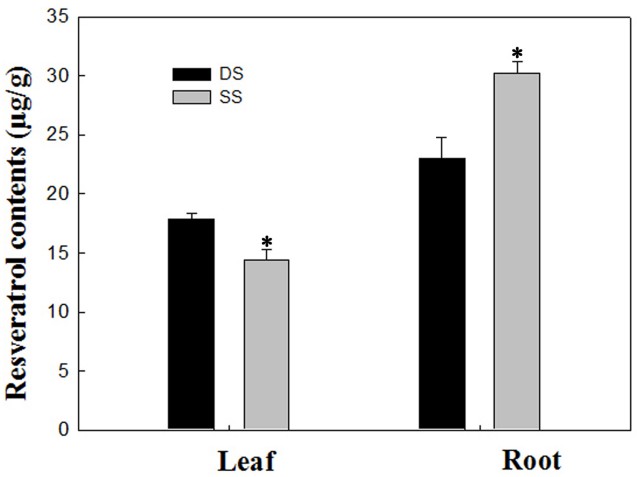

**Figure 7 Determination of resveratrol contents according to the method detailed by *Tang et al. (2010)* in leaf and root between SS and DS treatments.** Asterisks indicate significant differences from DS (*, $P < 0.05$).

## Peanut pod yields and associated genes under the SS sowing pattern

The number of pods and pod weight per plant is the direct factors affecting peanut yield. In this study, yield estimation indicated that the number of pods, full pods, and double kernels per plant under SS was higher than under DS. The theoretical pod yield of SS reached 1.225 kg, which was 12.9% higher than that of DS (1.085) (Table 4). Hormones, such as indole-3-acetic acid (IAA), abscisic acid (ABA), gibberellin (GA) and brassinosteroid (BR), play important roles in the regulation of growth and development in plants (*Tian et al., 2017*). Auxin response factor (ARF), a TF involved in auxin and regulating plant root growth and seed development (*Salmon, Ramos & Callis, 2008*), showed increased expression levels under SS. Nine genes that participate in auxin and BR biosynthesis were abundantly expressed under SS compared with under DS (Table 3). Two lipoxygenase family genes also had increased expression levels under SS.

## DISCUSSION

In recent years, with the rapid development of sequencing technology, genomes and transcriptomes have been analyzed to explore the mechanisms mediating various problems in agricultural production. However, few studies have focused on the mechanisms by which SS precision sowing improves yields in erect-plant-type peanut cultivars. Aerial flowering combined with underground fruit development are special characteristics of peanut plants; thus, the growth and development of root tissues are particularly important for peanut yields. Robust roots are the basis of crop growth and high yields (*Zheng et al., 2013*).

Various biotic and abiotic stresses under adverse environmental conditions severely reduce global crop production and food security (*Mengiste et al., 2003*). Peanut is one of the most susceptible crops to *Aspergillus flavus*. Abiotic stresses such as salinity, heat, mechanical damage, and drought seriously affect peanut crops' growth and development
**Table 3 List of putative candidate genes for high yields in SS sowing treatment.**

| Gene_id | Gene description | SR vs. DR |
|---|---|---|
| Oxidation-reduction | | |
| 107460187 | L-ascorbate oxidase homolog | 1.4 |
| 110279516 | L-ascorbate oxidase | 1.4 |
| 107482474 | L-ascorbate oxidase | 1.2 |
| 107480398 | L-ascorbate oxidase | 1.1 |
| 107471312 | Glutaredoxin | 2.8 |
| 107475918 | Glutaredoxin 3 | 2.4 |
| 107494759 | Glutaredoxin-C1-like | 1.5 |
| 107461150 | Glutaredoxin 3 | 1.1 |
| 107475477 | Cytochrome P450 83B1-like | 2.2 |
| 107491506 | Cytochrome P450 71A1-like | 2.1 |
| 107475482 | Cytochrome P450 83B1-like | 2.0 |
| 107475708 | Cytochrome P450 83B1-like | 3.5 |
| 107478833 | Cytochrome P450 84A1-like | 1.1 |
| 107475710 | Cytochrome P450 71A1-like | 1.2 |
| 107487896 | Cytochrome P450 71D8-like | 1.2 |
| 107492914 | Cytochrome P450 93A3 | 1.3 |
| Hormone related | | |
| 107491535 | Protein brassinosteroid | 1.7 |
| 107486286 | Gibberellin-regulated protein | 2.4 |
| BGI_novel_G000490 | Gibberellin receptor GID1 | 2.0 |
| 107495262 | Gibberellin receptor GID1 | 2.6 |
| 107476131 | Gibberellin receptor GID1 | 1.9 |
| 107464975 | Gibberellin 2-beta-dioxygenase 2-like | 1.7 |
| 107477445 | DELLA protein | 1.7 |
| 107491113 | DELLA protein | 1.1 |
| 107476118 | Auxin responsive GH3 gene family | 1.9 |
| 107465048 | Auxin-responsive protein IAA | 1.8 |
| 107492413 | Auxin response factor | 1.9 |
| 107463844 | Auxin-responsive protein IAA | 1.4 |
| 107462192 | Auxin responsive GH3 gene family | 1.8 |
| 107459619 | Auxin-responsive protein IAA | 1.5 |
| 107478268 | Auxin responsive GH3 gene family | 1.0 |
| 107464096 | Auxin-responsive protein IAA | 1.2 |
| 107495645 | Lipoxygenase | 2.4 |
| 107464479 | Linoleate 9S-lipoxygenase | 1.1 |
| Transcription factor and signaling regulation | | |
| 107491013 | MADS-box transcription factor | 3.0 |
| 107476111 | MADS-box transcription factor | 5.7 |
| 107458618 | Zinc finger protein CONSTANS-LIKE 4-like | 2.8 |
| 107492404 | Zinc finger protein | 2.4 |

| Gene_id | Gene description | SR vs. DR |
|---|---|---|
| 110279598 | Zinc finger protein 3-like | 2.3 |
| 107493757 | Zinc finger protein-like protein | 1.8 |
| 107481180 | Zinc finger protein 6 | 1.5 |
| 107460097 | C2H2-like zinc finger protein | 1.3 |
| 107483222 | MYB family transcription factor | 4.8 |
| 107461398 | MYB86 Transcription factor | 1.5 |
| 107475194 | MYB transcription factor MYB51 | 1.2 |
| 107491849 | R2R3 MYB protein 2 | 1.2 |
| 107459300 | WRKY transcription factor 70 | 2.4 |
| 107472118 | WRKY transcription factor | 1.8 |
| 107472768 | WRKY transcription factor 22-like | 1.5 |
| 107481590 | WRKY transcription factor 14 | 1.3 |
| 107463444 | ERF114-like | 2.4 |
| 107488356 | ERF022 | 1.6 |
| 107464842 | ERF13 | 1.5 |
| 107471058 | ERF WRI1 | 1.0 |
| 107486141 | ERF 1-like | 1.0 |
| BGI_novel_G002321 | bZIP | 7.3 |
| 107487341 | bZIP | 1.5 |
| 107466101 | Calcium-binding protein CML | 1.4 |
| 107491924 | Calcium-binding protein KIC-like | 1.3 |
| 107460313 | Calcium-binding protein CML | 1.2 |
| 107479434 | MAPKKK | 1.5 |
| **Disease resistance** | | |
| 107468687 | Pathogenesis-related protein 1 | 2.2 |
| 107468499 | Pathogenesis-related protein 1 | 1.8 |
| 107468493 | Pathogenesis-related protein 1 | 1.4 |
| 107474846 | Pathogenesis-related protein 1 | 1.0 |
| 107460041 | 4-coumarate–CoA ligase | 4.2 |
| 107472893 | 4-coumarate–CoA ligase | 3.1 |
| 107480465 | 4-coumarate–CoA ligase | 2.6 |
| 107458085 | 4-coumarate–CoA ligase | 2.0 |
| 107466660 | 4-coumarate–CoA ligase-like 5 | 1.7 |
| 107481001 | 4-coumarate–CoA ligase | 1.6 |

**Note:**
These candidate genes were up-regulated in SR vs DR comparison groups with the corrected $p$ value < 0.05 and are here classified according to predicted gene function. The values in the columns are the log2 Fold Change values for the SR vs DR comparison groups obtained from the transcriptome results. SR, root in single-seed (SS) precision sowing; DR, root in double-seed (DS) sowing.

(*Amin et al., 2019*). In our study, families of TFs, including Zinc finger protein, MYB, bZIP, ERF and WRKY, were increased in the SS sowing treatment compared with the DS sowing treatment. Many studies have identified these TFs as being involved in abiotic stress

**Table 4 Pod per plant at maturity stage of peanut under different planting patterns.** Values followed by different lowercase letters indicate significant difference at $P \leq 0.05$.

| Treatment | Pods number per plant | Full pods number per plant | Double kernel number per plant | Pod weight per plant (g) | Economic coefficient |
|---|---|---|---|---|---|
| SS | 39.1a | 22.6a | 24.4a | 51.7a | 0.49a |
| DS | 31.8b | 16.3b | 16.9b | 39.3b | 0.45b |

responses (*Wei et al., 2017*; *Wang et al., 2016*; *Banerjee & Roychoudhury, 2015*). In addition, secondary messengers, like $Ca^{2+}$ and ROS, trigger signaling proteins downstream, such as calcium-dependent protein kinases (CDPKs), calmodulin (CaM), calcineurin-B-like proteins (CBLs), mitogen-activated protein kinase (MAPK) cascades and ROS-modulated protein kinases (PKs), and these can also respond to numerous plant development and environmental challenges (*Yang et al., 2019*). Genes encoded calcium-binding protein CML, calcium-binding protein KIC-like, and MAPKKK, which is part of the $Ca^{2+}$ signaling pathway, were expressed at higher levels under SS (Table 3). Genes involved in the MAPK cascade include interlinked MAPK, MAPKK, and MAPKKK, and such cascades play important roles in signal transduction of plant hormones, biotic stresses, and abiotic stresses (*Wang et al., 2017*).

Resveratrol, an important phytoalexin, can be induced by pathogenic bacteria or other stimuli and has strong bactericidal and defensive activity in plants. So far, the heterogeneous transformation of Res genes from rice, barley, wheat, tomato, tobacco and other crops has been reported. All these transgenic crops exhibit improved resistance to diseases such as rice blast (*Stark, Nelke & Hanbler, 1997*), powdery mildew in barley and wheat (*Fettig & Hess, 1999*) and gray mold in tomato and tobacco (*Hain, Bieseler & Kindl, 1990*). The rate of Res induction in peanut seeds has also been linked to resistance to *Aspergillus flavus* infection (*Fajardo et al., 1994*). Res is synthesized by the phenylalanine metabolic pathway in plants, which includes four key enzymes: cinnamate-4-hydroxylase (C4H), phenylalanine ammonia lyase (PAL), 4-coumarate-CoA ligase (4CL), and STS. STS genes can be divided into two types, one of which is resveratrol synthase (RS), which synthesizes Res using malonyl-CoA and coumaric acid-CoA as substrates, and this type mainly exists in *Arachis hypogaea* and *Vitis vinifera*. The other is pinosylvin synthase (PS), which utilizes malonyl-CoA and cinnamyl-CoA as substrates, and has mainly been identified in *Pinus sylvestris* and *Pinus strobus*. RS is the major rate-limiting enzyme for resveratrol synthesis, which is reported to participate in ROS resistance and is a key element in stress resistance processes (*Chang et al., 2011*). Endogenous or exogenous Res can also reduce ROS content in plants (*Zheng et al., 2015*). These Res's effects are likely to promote disease resistance and antioxidant ability in peanut roots, thus supporting high-quality harvests and high yields of peanuts.

In this study, high activities of the enzymes POD, SOD and CAT in the ROS scavenging pathway were observed in roots of plants grown under SS (Table S1). SOD, as a key element of enzymatic defense systems, catalyzes the disproportionation of radicals $O_2^{\bullet-}$

to $O_2$ and $H_2O_2$, the latter of which can then be scavenged by POD or CAT. The elevated activity of antioxidant enzymes can be regarded as an effective mechanism for resisting oxidative stress. The balance of the redox state of plant cells contributes to improving plant resistance to abiotic stress (*Duan et al., 2012*). It has been reported that overexpression of glutaredoxin in tomatoes can confer drought, oxidation, and salt resistance (*Guo et al., 2010*). These results suggested oxidoreductases are involved in protecting plants from various abiotic stresses through maintaining oxidation-reduction homeostasis and scavenging surplus ROS. CYP catalyzes the biological oxidation of various substrates through molecular oxygen activation and acts as a pivotal part of stress responses and metabolic processes, exhibited as differential transcription levels under SS and DS treatments (*Xiong et al., 2017*).

Based on previous studies, auxin and BR are known to play important roles in root development. Still, the root tip phenotypes of auxin mutants differ from those of BR mutants (*González-García et al., 2011*), indicating that the auxin and BR pathways acting on the root are not the same. In *Arabidopsis*, the BR encoding gene *AtDWF4* was found to regulate leaf growth by promoting cell expansion (*Hur et al., 2015*). Its overexpression in *Brassica napus* can increase seed production (*Sahni et al., 2016*). Many unigenes encoding auxins and BR biosynthesis-related genes are expressed at a high level under SS. Lipoxygenase, which exhibited different expression levels between SS and DS treatments, is an enzyme that is important to the synthesis of jasmonic acid, which plays a vital role in stored lipid migration during seed germination (*Rahimi et al., 2016*). The up-regulated expression of lipoxygenases under SS is likely to provide sufficient nutrients essential for pod development. ABA also plays crucial roles in seed dormancy and development (*Chauffour et al., 2019*). These results strongly suggested that the biotic and abiotic stress tolerance in roots of peanut plants grown from SS precision sowing were enhanced compared with those under the DS sowing treatment. These mechanisms may explain the higher yields achieved under SS precision sowing based on patterns at the transcriptional level.

## CONCLUSIONS

This study performed a comparative transcriptomic analysis of peanut leaves and roots between precision SS sowing and standard DS sowing treatments. Genes involved in resveratrol synthesis were found to be dramatically induced in roots under the SS treatment. Accordingly, a higher content of Res was detected under SS precision sowing. In addition, genes involved in calcium signaling transduction and phytohormone metabolism were identified to be differentially expressed in SS and DS roots. Redox process genes and TFs, including WRKY, MYB, MADS-box and zinc finger proteins, were up-regulated in roots of plants grown under the SS precision sowing treatment. These genes participate in resistance to biotic and abiotic stresses, thus protecting plants from disease and drought, salinity, and chilling stress. These results may inform breeding to enhance roots' activity and growth, providing a basis for achieving higher yields in peanuts. To the best of our knowledge, we have provided a novel mechanism by which SS precision sowing improves erect-plant-type peanut yields.

## LIST OF ABBREVIATIONS

| | |
|---|---|
| **ABA** | Abscisic acid |
| **BR** | Brassinosteroid |
| **ARF** | Auxin response factor |
| **CAT** | Catalase |
| **CBLs** | Calcineurin-B-like proteins |
| **CDPKs** | Calcium-dependent protein kinases |
| **C4H** | Cinnamate-4-hydroxylase |
| **CIPK** | CBL-interacting protein kinase |
| **4CL** | 4-coumarate-CoA ligase |
| **CYP** | Cytochrome P450 |
| **DEGs** | Differentially expressed genes |
| **FPKM** | Fragments per kb per million |
| **GA** | Gibberellin |
| **GATK** | Genome analysis toolkit |
| **GO** | Gene ontology |
| **HISAT** | Hierarchical indexing for spliced alignment of transcripts |
| **IAA** | Indole-3-acetic acid |
| **INDEL** | Insertion–Deletion |
| **MAPK** | Mitogen-activated protein kinase |
| **NBT** | Nitroblue tetrazole |
| **PAL** | Phenylalanine ammonia-lyase |
| **POD** | Peroxidase |
| **PS** | Pinosylvin synthase |
| **qRT-PCR** | Quantitative real-time PCR |
| **Res** | Resveratrol |
| **RSEM** | RNA-Seq by expectation-maximization |
| **SNP** | Single nucleotide polymorphisms |
| **SOD** | Superoxide dismutase |
| **STS** | Stilbene synthase |
| **WGCNA** | Weighted gene correlation network analysis |

### Funding

This work was supported by the following grants: National Key R&D Program of China (2018YFD1000900); Major Basic Research Project of Natural Science Foundation of Shandong Province (2018GHZ007); Science and Technology Innovation Project of Shandong Academy of Agricultural Sciences (CXGC2018D04); Agricultural scientific and technological innovation project of Shandong Academy of Agricultural Sciences

(CXGC2018E13). The funders had no role in study design, data collection and analysis, decision to publish, or preparation of the manuscript.

## Grant Disclosures

The following grant information was disclosed by the authors:
National Key R&D Program of China: 2018YFD1000900.
Natural Science Foundation of Shandong Province: 2018GHZ007.
Shandong Academy of Agricultural Sciences: CXGC2018D04 and CXGC2018E13.

## Competing Interests

The authors declare that they have no competing interests.

## Author Contributions

- Sha Yang performed the experiments, prepared figures and/or tables, authored or reviewed drafts of the paper, and approved the final draft.
- Jialei Zhang performed the experiments, prepared figures and/or tables, authored or reviewed drafts of the paper, and approved the final draft.
- Yun Geng performed the experiments, prepared figures and/or tables, and approved the final draft.
- Zhaohui Tang performed the experiments, authored or reviewed drafts of the paper, and approved the final draft.
- Jianguo Wang performed the experiments, authored or reviewed drafts of the paper, and approved the final draft.
- Feng Guo performed the experiments, authored or reviewed drafts of the paper, and approved the final draft.
- Jingjing Meng analyzed the data, prepared figures and/or tables, tended the plants, and approved the final draft.
- Quan Wang analyzed the data, prepared figures and/or tables, tended the plants, and approved the final draft.
- Shubo Wan conceived and designed the experiments, authored or reviewed drafts of the paper, and approved the final draft.
- Xinguo Li conceived and designed the experiments, authored or reviewed drafts of the paper, and approved the final draft.

## Data Availability

The RNA-seq raw sequencing data are available in the SRA database: PRJNA497502 (SRA accession number: SRP166140). qRT-PCR data are available as Supplemental Files.

## Supplemental Information

Supplemental information for this article can be found online at http://dx.doi.org/10.7717/peerj.10616#supplemental-information.

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
