# Peer review of "Transcriptome analysis reveals the mechanism of improving erect-plant-type peanut yield by single-seeding precision sowing"

_PeerJ, doi:10.7717/peerj.10616_

## Round 0.1 · original submission · Major Revisions

Although the research is interesting and applicable for the scientific community, the introduction and figure/table descriptions need major revision.

The introduction does not tie cited research to specific project aims. Correlations between what has been done and referenced vs. what the researchers aimed to accomplish have not been made. The purpose of the research, hypotheses, methods, and assessment of results has not be included in the introduction.

Reviewer 1 ·

Basic reporting

No comment.

Experimental design

RNA sequencing experiments should contain three biological repeats for each treated sample.Data from only one test is not appropriated for publication, but could work as a hint of the research.

Validity of the findings

The transcriptomic data can provide a possibilty of the biological question.In the manucript, the authors found that the expression of resveratrol synthesis genes were induced in peanut roots by single-seed precision sowing and they determined the level of resveratrol. These findings showed that the resveratrol synthesis was enhanced in the roots by single-seed precision sowing. However, whether its enhance is associated with the germinaltion and growth of peanuts remain unclear. Also, resveratrol contributes to stress of adaptation of plants needs solid evedence.

Additional comments

The manuscript described a comparative transcriptomic results. The auhours found that the expression of resveratrol synthesis genes were induced in peanut roots by single-seed precision sowing and the level ofresveratrol was increased in the roots with the same treatment. It may not the mechanism of the increase of peanut germination and growth.
1) The bar is missed in Fig. 1a, and a statistic test is lacking in Fig.1b.
2)The error bars were lost in Fig 2C(?)。
3)Fig. 3a can be shown as a bar chart.
4) Words in Fig. 5 were not clear.
5) No statistic test data in Fig 7.

·

Basic reporting

Intro and Background-
1. In the abstract, a statement about the relationship of resveratrol to stress is needed. The authors should tie in the role of resveratrol in managing plant stress, lines 75 and 76.
2. Consider comparing typical sowing methods in China to practices in the US and other peanut growing regions, lines 92 and ff in the introduction. This would give context and wider application to the scientific community at large.
3. The paragraph describing stress and previous research is lacking in fully developing the relationship between previous research, genes and methods, lines 103-109. This entire section should be revised and improved upon.
4. On line 109, a new section describing the importance of aflatoxin contamination and resveratrol is included. I would suggest beginning a new paragraph here especially if the first part of the section is more detailed.
5. The authors should tie in the importance of resveratrol in aflatoxin reduction or leave that out. Clarification is needed, lines 109-113.
6. Qualify “higher content”, line 116.
7. Clarify importance of gene discussion to project aims, line 127.
8. At the conclusion of the intro and background, the authors should describe the purpose of the research, hypothesize about the findings of increased expression of genes to SS planting efficiency and plant viability, and describe how these findings will be used to assess results, after line 148. This is of primary importance in communicating the scientific findings of the research.

Figures-
For most of the figures and tables included in the packet, there is insufficient information. For example:
1. Figure 1a- which picture is from a SS treatment and which is DS? There should be a description of how the pictures relate to the data shown in part b.
2. Figure 4- there should be a legend for the colors showing in part a. How do the colors relate to quantity? The gene ID numbers should be cross-referenced with table 3 and use gene descriptors. Part b does not show units or provide any information about the number of samples/replicates or what the numbers represent.
3. Figure 6b- What do the clusters represent? Is there a color key, i.e. resveratrol gene pathways blue cluster? Do the colors from a and b correlate? More information is needed to clarify importance of the results.
4. Figure 7- need to include N and method of detection in the description.
5. Table 1- include a key for the abbreviations used in the table i.e., SL, DL etc.
6. Table 2- need a reference for the method used to calculate ratio.
7. Table 3- need a reference for the gene identity database.
8. Table 4- the right-hand section appears to be cut off, economic what?

Grammar and English Usage
1. Use “precision” as opposed to “precise” throughout the paper and in the title, lines 97, 98 and 358. There are probably other instances so I would recommend a search for precise in the body of the document.
2. Line 93: Replace “The” with “A”
3. Line 94: add “of” after tons.
4. Line 95: annual yield of peanuts, crops, or this particular farm?
5. Line 98: Add “the” or “a” before conventional.
6. Line 104: are “growth” and “development” unnecessary synonyms?
7. Line 110: Does “peanut” need to be plural?
8. Line 137: Would replacing the comma with a colon and removing “namely” be better?
9. Lines 174, 179, 181: The instructions switch from impersonal to first person halfway through the paragraph. This could indicate some change in how they performed the experiments, but does this need to be changed to match the whole paragraph?
10. Line 357: “Encountered” can be removed.
11. Line 398: Is “Accordingly” needed here or could it be cut?
12. Lines 567-569: For this citation sources say "Journal of the Faculty of Science, Hokkaido Imperial University" is this the proper credit for Japanese publishing?

Experimental design

1. The abbreviation, KEGG, is mentioned several times throughout the paper including in the methods section, line 182. Include a statement on what the letters represent and the importance/significance of the pathway in evaluating results.
2. In the results section, there are several gene families called out as significant, lines 338-339 and 345-351. Either in the table or in the body of the text include gene id# so the data can be compared. Another option would be to include the log fold difference values in the body of the text in these sections.
3. The research question is not well defined or included in the introduction.
4. The significance of the research is not tied well to previous and cited references illustrating the need for the research project.

Validity of the findings

1. Statistical measurements and replicates should be included on all figures and tables.
2. Line 361: There is a statement about how robust roots are the basis of crop growth and high yields. Is there a reference or references to validate this statement? Is the case laid out for this measure in the introduction?

---

## Round 0.2 · Minor Revisions

Please provide a supplement with the sequences used in this study, as we can't find genes with BGI_novel or the numerical gene ID at the peanutbase.org site.

Reviewer 1 ·

Basic reporting

I have no question.

Experimental design

I have no question.

Validity of the findings

I have no question.

Additional comments

I have no question.

---

## Round 0.3 · Minor Revisions

Though the authors corrected the manuscript, it is still hard to find the detailed information of the reference genome for validation in future studies. Please provide the transcriptome sequence or at least the NCBI accession number in the revised version.

---

## Round 0.4 · Minor Revisions

The markup file that I attached was for version 3, and I am still seeing a version 3. I would expect to see a version 4 with suggested changes as indicated in the markup files. Please re-check your records to determine whether this has actually been revised.

---

## Round 0.5 · Major Revisions

The previous Academic Editor has been unresponsive so Ihave taken over the submission.

In the version (v4) sent back I see that some of the suggested edits that were attended to, but misinterpreted (someone familiar with language structure should have tended to it). In general, though a few places were marked it would be of value for someone language familiar to proofread the entire manuscript. It is not a goal to help us guess what you did, but actually in clear detail inform us. This version again touches on the areas which were changed; however, a general overview would be beneficial.

There were key areas in the data which did not add up or provide the reader with the needed information to interpret the findings. There is raw RNA sequencing data pointed to, but it by itself does not lead us to the path of the Gene_IDs or their sequence. There should be an appropriate FASTA file or a pointer to a database source where the sequence data can be located and correspond to the annotation terms which were provided. The information alluded to GeneIDs in supplemental table S2, but there is no connection to the actual sequence. Likewise, many are listed a novel without any representation.

In a supplement, it would also help to list the GO:12345 annotation alongside the textual annotations listed in Figure 5. In general additional data should be provided, or a clearer representation of the supplemental data provided. I again consider the manuscript requiring additional revisions.

Apologies for the long process, but it is not the role of the editor to proof the manuscript. It is best to be able to evaluate a clearly assembled manuscript with data; trying to wade through various details can cause unnecessary delays.

---

## Round 0.6 · accepted · Accept

It appears that you have attended to many of the concerns and I reluctantly will approve the changes. The data provided is very hard for the reader to try to obtain and review the data, but through a painstaking process can obtain a general retrieval of what are trying to present. Better organization that helps lead the reader into following and validating your assessments is a goal, but is demonstrated marginally in this case. I do not have great anticipation that the results will lead toward validating mechanisms that are important in highlighting the different growth patterns; however, I do hope that you can prove this assessment wrong. I will leave the readership to come to their own conclusions. I will accept this manuscript in its current form and allow it to move forward. Congratulations on your efforts.